# Curvature Tuning: Provable Training-free Model Steering From a Single Parameter

**Leyang Hu**[*]
Brown University
leyang_hu@brown.edu

**Matteo Gamba**[*]
KTH
mgamba@kth.se

**Randall Balestriero**
Brown University
randall_balestriero@brown.edu

## Abstract

The scaling of model and data sizes has reshaped the AI landscape, establishing finetuning pretrained models as the standard paradigm for solving downstream tasks. However, dominant finetuning methods typically rely on weight adaptation–thus often lacking interpretability– and depend on heuristically chosen hyperparameters. In this paper, we take a different perspective and shift the focus from weights to activation functions, viewing them through the lens of spline operators. We propose Curvature Tuning (CT), an interpretable and principled steering method that modulates a model's decision boundary by injecting a single hyperparameter into its activation functions. Making this hyperparameter trainable gives rise to a novel and highly parameter-efficient finetuning method. This perspective complements current finetuning methods–whose effect lies primarily in feature adaptation–empirically improving both generalization and robustness.

## 1 Introduction

The scaling of model and data sizes has fueled a paradigm shift in machine learning: transitioning from training task-specific models from scratch to finetuning pretrained foundation models to downstream applications. Full finetuning, the process of steering a pretrained model by adapting all its parameters to downstream datasets, was once the primary approach for transferring knowledge. While it effectively enhances generalization [1] and robustness [2], it is computationally expensive at large model scales. To mitigate this, parameter-efficient finetuning (PEFT) methods such as Serial Adapter [3] and LoRA [4] have been introduced, which finetune only a small subset of parameters. However, these approaches usually lack interpretability and principled design. For instance, they treat the model as a black box, making it unclear how the model is steered for downstream tasks. Typically, they rely on heuristic choices—such as LoRA's rank, placement, and initialization—with minimal theoretical guidance. This leads to a natural question: *how can we construct principled steering solutions addressing both efficiency and interpretability?* This work answers the question by introducing a novel perspective. We observe that despite differences in specific forms, existing finetuning methods all share a focus on adapting model weights. However, one critical model component has been largely overlooked: the activation functions (e.g., ReLU), which are responsible for the model's nonlinearity and, ultimately, its expressivity [5, 6].

**Contributions**. Grounded in the spline interpretation of deep networks [7, 8], (1) we propose **Curvature Tuning (CT)**, a steering method that provably modulates a model's decision boundary curvature by injecting a single hyperparameter $\beta$ into the activation function, as shown in Fig. 1. (2) Additionally, allowing $\beta$ to be trained leads to a novel finetuning method. (3) CT is **highly parameter-efficient**: as a steering method, it introduces only one (hyper)parameter per network. As a finetuning method, *Trainable CT* still uses significantly fewer parameters than LoRA with rank one, requiring only 0.58% to 59.05% of the parameters used by LoRA in our experiments.

---

[*]These authors contributed equally to this work.

39th Conference on Neural Information Processing Systems (NeurIPS 2025).

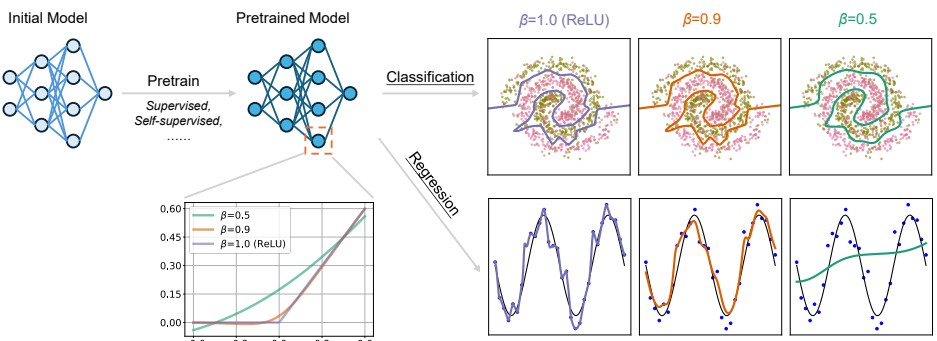

Figure 1: **Illustration of Curvature Tuning (CT)** on classification (top) and regression (bottom) tasks. **CT steers a pretrained model** by replacing ReLUs with a $\beta$-parameterized activation function and tuning $\beta$ from 1 to 0, **effectively modulating the model's decision boundary curvature.**

## 2 Background

Finetuning refers to steering[1] a pretrained model to improve its downstream performance. Initially, the common practice was to continue training all model parameters (*full finetuning*). However, with growing model scales, the practice has become increasingly costly, especially given the limited size of many downstream datasets, giving rise to *parameter-efficient finetuning (PEFT)*. **Additive PEFT** adds trainable parameters to the pretrained model, adapting only these new parameters during finetuning. Examples include Serial Adapter [3], Prefix-tuning [9], (IA)[3] [10] and RoAd [11]. **Selective PEFT** identifies a subset of parameters for finetuning, as in U-Diff and S-Diff pruning [12]. **Reparameterized PEFT** decomposes pretrained weights into low-rank matrices, finetuning the low-rank components, which are converted back during inference; examples include LoRA [4] and DyLoRA [13]. **Hybrid PEFT** combines multiple PEFT approaches [14, 15]. While PEFT methods differ in the parameters they update, they all adapt model weights and operate on learned features—an approach that often relies on heuristic tuning. In contrast (Section 3), *CT* introduces only a single hyperparameter in the activation functions by modulating curvature, offering a more interpretable alternative that operates on the model's underlying function space, without changing model weights.

## 3 Curvature Tuning (CT): a provable method for model steering

**The spline formulation of deep networks** We begin by briefly introducing relevant concepts in spline theory, which provide a mathematical framework for CT. A *spline function* is a continuous function $s : \mathbb{R}^D \to \mathbb{R}$ defined piecewise by polynomials. An *affine spline function* is a special case where each piece is defined by an affine mapping. Such a function can be parameterized by three components: a matrix $\mathbf{A} \in \mathbb{R}^{R \times D}$ representing the slopes of the affine mappings, a vector $\mathbf{b} \in \mathbb{R}^R$ representing the offsets, and a partition $\Omega \triangleq \{\omega_1, \ldots, \omega_R\}$ of the input space $\mathbb{R}^D$ into $R$ regions. For an input $\mathbf{x} \in \mathbb{R}^D$, the affine spline function is defined as $s[\mathbf{A}, \mathbf{b}, \Omega](\mathbf{x}) = \sum_{r=1}^{R} (\langle \mathbf{A}_{r,\cdot}, \mathbf{x} \rangle + \mathbf{b}_r) \mathbf{1}_{\{\mathbf{x} \in \omega_r\}}$, where the indicator function $\mathbf{1}_{\{\mathbf{x} \in \omega_r\}}$ equals 1 if $\mathbf{x}$ belongs to region $\omega_r$ and 0 otherwise. The key result underpinning our study is that many deep network layers—such as fully connected and convolutional, and convex piecewise-linear activations (e.g., ReLU, max pooling, or maxout)—can be exactly represented as *max-affine spline functions* [8] (further details in Appendix A), which are special affine splines that do not need explicit knowledge of $\Omega$:

$$s[\mathbf{A}, \mathbf{b}](\mathbf{x}) = \max_{r=1\ldots R} (\langle \mathbf{A}_{r,\cdot}, \mathbf{x} \rangle + \mathbf{b}_r) \tag{1}$$

$$= \sum_{r=1}^{R} \mathbf{t}_r (\langle \mathbf{A}_{r,\cdot}, \mathbf{x} \rangle + \mathbf{b}_r) \tag{2}$$

for one-hot encoded selection variable $\mathbf{t} \in \{0, 1\}^R$, with non-zero component $\mathbf{t}_{r^*} = 1$, for $r^* = \arg\max_{r=1,\ldots,R} (\langle \mathbf{A}_{r,\cdot}, \mathbf{x} \rangle + \mathbf{b}_r)$.

---

[1]We use *steering* as a general term for model tuning, while *finetuning* for training-based parameter adaptation.

In the following, we construct a model steering method operating on the activation functions, thereby changing their local curvature, without modifying the network's weights. For each neural network layer interpretable as a max-affine spline operator (Eq. (1)), the method acts by smoothing the neuron's nonlinearity. This can be done in two ways, as detailed below.

**1. Smoothing the spline region assignment process** In Eq. (2), the affine transformation is selected in a *hard* manner, picking the region index maximizing the activation output. Alternatively, the variable $\mathbf{t}$ can be inferred via the following regularized (*soft*) region selection problem [16]:

$$\mathbf{t}^{\beta} = \arg \max_{\mathbf{t} \in \Delta_R} \left[ \beta \sum_{r=1}^{R} \mathbf{t}_r \cdot (\langle \mathbf{A}_{r,\cdot}, \mathbf{x} \rangle + \mathbf{b}_r) + (1 - \beta) H(\mathbf{t}) \right], \qquad (3)$$

where $H(\mathbf{t})$ denotes the Shannon entropy of the selection variable, and $\Delta_R$ is the probability simplex.

**2. Smoothing the max computation** Instead of soft region assignment, we can instead directly smooth the maximum function in Eq. (1), leading to the log-sum-exp operator (i.e. SoftPlus):

$$(1 - \beta) \ln \left[ \sum_{r=1}^{R} \exp \left( \frac{\langle \mathbf{A}_{r,\cdot}, \mathbf{x} \rangle + \mathbf{b}_r}{1 - \beta} \right) \right], \qquad (4)$$

where $\beta \to 1$ recovers the original affine spline activation, e.g., ReLU.

**Implementation of CT** By combining the soft parameterizations in Eq. 3 and 4, we introduce an expressive activation function, which we name **CT Unit (CTU)**:

$$\varphi_{\beta,c}(\mathbf{x}) = c \cdot \sigma \left( \frac{\beta \mathbf{x}}{1 - \beta} \right) \cdot \mathbf{x} + (1 - c) \cdot \ln \left[ 1 + \exp \left( \frac{\mathbf{x}}{1 - \beta} \right) \right] \cdot (1 - \beta), \qquad (5)$$

where $\beta \in [0, 1]$ modulates the curvature, $c \in [0, 1]$ is the mixing coefficient, and $\sigma(\cdot)$ denotes the sigmoid function. This is essentially a convex combination of reparameterized SiLU and SoftPlus:

$$\text{SiLU}(\mathbf{x}) = \sigma(\eta \mathbf{x}) \cdot \mathbf{x}, \ \eta = \frac{\beta}{1 - \beta}; \quad \text{SoftPlus}(\mathbf{x}) = \frac{1}{\gamma} \cdot \ln \left[ 1 + \exp (\gamma \mathbf{x}) \right], \ \gamma = \frac{1}{1 - \beta}. \qquad (6)$$

Sec. A expands upon the theoretical motivation of CTU, while Sec. C provides a theoretical interpretation of its shaping of curvature. Importantly, combining the two soft parameterizations yields an expressive activation function, encompassing activations such as ReLU, SiLU, SoftPlus, and GELU.

**Steering vs Trainable CT**. We conclude by providing two implementations of CT differing in how CTU is applied. The first, denoted *CT*, replaces all ReLUs in the network with CTUs using fixed $c = 0.5$ and a shared $\beta \in [0, 1]$. This version is highly parameter-efficient—introducing only a single hyperparameter—and does not require backpropagation, making it suitable as a steering method. The second, named *Trainable CT*, also replaces all ReLUs with CTUs but assigns each output neuron its own trainable pair $(\beta, c)$, optimized via backpropagation. This version serves as a finetuning method: while it introduces additional parameters, the increase is modest compared to methods like LoRA.

## 4 Enhancing Model Generalization and Robustness with CT

**Improving generalization on downstream datasets.** We evaluate the effectiveness of *CT* and *Trainable CT* in improving model generalization across a variety of downstream datasets. Specifically, we transfer ImageNet-pretrained ResNet-18/50/152 models to 12 downstream datasets (details in Appendix B.1). For comparison, we consider two baselines: (i) linear probing on the pretrained backbone, and (ii) finetuning the backbone with LoRA (rank $r = 1$, scale $\alpha = 1$) while training the linear head (experimental details in Appendix B.1 and Appendix B.2).

Results in Table 1 and Table 3 show that *CT* improves generalization compared to linear probing, with average relative gains of 1.97%/1.16%/0.02% on ResNet-18/50/152. *Trainable CT* achieves the highest performance across all methods, with average relative improvements on ResNet-18/50/152 of 6.75%/8.59%/8.34% over linear probing; 4.62%/7.14%/8.51% over *CT*; and 10.20%/4.64%/1.70% over LoRA. Importantly, *Trainable CT* achieves better performance than LoRA with far fewer parameters. On ResNet-18/50/152, the number of trainable parameters (excluding the classifier) is only 11.05%/57.20%/59.09% of that required by LoRA, even when LoRA is set to its lowest rank

Table 1: Accuracy (%) of ImageNet-pretrained ResNet-18/50 when transferred to 12 downstream datasets. Inside the parentheses are number of trainable parameters (excluding the linear classifier).

| | ResNet-18 | | | | ResNet-50 | | | |
| Dataset | Frozen (0) | CT (1) | LoRA (35923) | Train CT (3968) | Frozen (0) | CT (1) | LoRA (79443) | Train CT (45440) |
|---|---|---|---|---|---|---|---|---|
| Arabic Characters | 81.91 | 87.65 | 93.37 | **93.76** | 80.65 | 83.66 | 94.21 | **95.67** |
| Arabic Digits | 97.93 | 98.77 | **99.08** | 99.03 | 98.33 | 98.37 | 99.08 | **99.16** |
| Beans | 87.76 | 90.36 | 93.23 | **94.01** | 89.58 | 91.93 | 94.79 | **95.57** |
| CUB-200 | 62.84 | 63.18 | 54.83 | **64.30** | 65.23 | 64.62 | 66.17 | **71.03** |
| DTD | 62.80 | 62.66 | 54.36 | **63.62** | **67.34** | 66.91 | 64.70 | 65.07 |
| FashionMNIST | 88.63 | 88.70 | **91.65** | 91.07 | 90.05 | 90.34 | 92.19 | **92.78** |
| FGVC-Aircraft | 36.80 | 38.68 | 29.19 | **46.44** | 38.03 | 41.16 | 41.99 | **55.70** |
| Flowers102 | 80.86 | 81.97 | 67.53 | **86.55** | 84.00 | 83.84 | 82.58 | **87.62** |
| Food101 | 61.41 | 62.27 | 64.40 | **66.04** | 68.06 | 68.02 | 71.42 | **73.60** |
| DermaMNIST | 74.83 | 75.05 | 74.21 | **77.66** | 75.94 | 75.89 | 75.73 | **78.02** |
| OCTMNIST | 65.03 | 67.27 | **74.27** | 69.53 | 67.53 | 68.00 | **75.90** | 74.13 |
| PathMNIST | 86.77 | 87.51 | **87.62** | 87.17 | 90.08 | **90.26** | 85.43 | 87.33 |
| Average | 73.96 | 75.34 | 73.64 | **78.26** | 76.24 | 76.92 | 78.68 | **81.31** |

($r = 1$), underscoring the parameter efficiency of CT. Additional experiments demonstrating the effectiveness of *Trainable CT* on transformers is provided in Appendix B.3.

**Improving robustness to adversarial and corrupted data.** To conclude, we demonstrate that *CT* can enhance model robustness **without any adversarial training**. We evaluate robustness of ResNet-18/50/152 on CIFAR-10/100 and ImageNet using the $\ell_2/\ell_\infty$/corruption benchmarks from RobustBench [17]. Here, when applying *CT*, we replace all ReLU activations in the backbone with CTUs and perform a grid search over $\beta \in [0.7, 1]$ with a step size of 0.01, reporting the value that yields the best performance on each benchmark. For experimental details, see Appendix B.4. As summarized in Table 2, *CT* is particularly effective against $\ell_\infty$ attacks, achieving large relative improvements of 44.01%/1032.64%/1494.46% for ResNet-18/50/152. We also show that *Trainable CT* can also enhance the model's $\ell_\infty$ robustness without adversarial training in Appendix B.5.

Table 2: Robust accuracy (%) of ImageNet-pretrained ResNets under $\ell_2/\ell_\infty$ attacks and corruptions.

| | | $\ell_2$ | | | $\ell_\infty$ | | | Corruption | | |
| Model | Dataset | Base | CT | $\beta$ | Base | CT | $\beta$ | Base | CT | $\beta$ |
|---|---|---|---|---|---|---|---|---|---|---|
| | CIFAR10 | 53.67 | 53.67 | 1.00 | 11.17 | **14.93** | 0.90 | 77.73 | 77.73 | 1.00 |
| ResNet18 | CIFAR100 | 24.30 | **25.50** | 0.92 | 4.47 | **6.90** | 0.92 | 51.81 | **51.95** | 0.94 |
| | ImageNet | 23.37 | 23.37 | 1.00 | 0.00 | **7.00** | 0.89 | 33.11 | **33.32** | 0.92 |
| | Average | 33.78 | **34.18** | 0.97 | 5.21 | **9.61** | 0.90 | 54.22 | **54.33** | 0.95 |
| | CIFAR10 | 55.10 | **56.53** | 0.97 | 10.10 | **12.08** | 0.90 | 77.26 | 77.26 | 1.00 |
| ResNet50 | CIFAR100 | 23.83 | **25.80** | 0.96 | 4.43 | **7.90** | 0.93 | 53.91 | **53.93** | 0.98 |
| | ImageNet | 31.90 | 31.90 | 1.00 | 0.30 | **9.30** | 0.93 | 39.64 | 39.64 | 1.00 |
| | Average | 36.94 | **38.08** | 0.98 | 4.94 | **10.68** | 0.94 | 56.94 | 56.94 | 0.99 |
| | CIFAR10 | 56.27 | 56.27 | 1.00 | 11.47 | **15.00** | 0.99 | 78.82 | **78.83** | 0.99 |
| ResNet152 | CIFAR100 | 27.90 | **28.23** | 0.98 | 5.40 | **7.70** | 0.99 | 56.12 | 56.12 | 1.00 |
| | ImageNet | 42.50 | 42.50 | 1.00 | 0.30 | **13.53** | 0.97 | 45.47 | 45.47 | 0.99 |
| | Average | 42.22 | **42.33** | 0.99 | 5.72 | **12.08** | 0.98 | 60.14 | 60.14 | 0.99 |

## 5   Conclusion

This paper proposes Curvature Tuning (CT), an interpretable and principled model steering method that provably modulates a model's decision boundary via a single parameter injected into its activation functions, without changing the model weights. We apply CT in two forms: as a steering method with fixed parameters (*CT*) and as a finetuning method with learnable ones (*Trainable CT*). Both improve generalization and enhance robustness, with *Trainable CT* approaching LoRA's performance.

## Acknowledgments and Disclosure of Funding

Computations were in part enabled by the Berzelius resource provided by the Knut and Alice Wallenberg Foundation at the National Supercomputer Centre.

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

# Appendix

The remainder of the paper collects additional experimental validation and theoretical derivations supporting our main results. The appendix is organized as follows.

1. Appendix A briefly connects several deep network architectures to affine spline operators.
2. Appendix B details our experimental setup.
3. Appendix C provides theoretical intuition behind *CT*.
4. Appendix D provides pseudocode for *CT* as well as *Trainable CT*.
5. Appendix E provides pseudocode for LoRA, describing how the method was applied throughout our experiments (Section 4).

## A   Spline Theory

The spline theory of deep learning establishes that a large class of deep network (DN) layers can be modeled as Max Affine Spline Operators (MASOs). More precisely:

**Theorem A.1.** *(Propositions 1-4 in Balestriero and Baraniuk [8]) Any DN layer comprising a linear operator (e.g., fully connected or convolutional layer) followed by a convex and piecewise affine non-linear operator (e.g., ReLU, leaky-ReLU, absolute value activation, max/average/channel pooling, maxout; with or without skip connections) is a MASO.*

Consequently, a deep network (e.g., MLP, CNN, RNN, ResNet) composed of such linear operators and convex, piecewise affine non-linear operators is a composition of MASOs. However, it is important to note that the network as a whole is not a MASO but an Affine Spline Operator (ASO). In other words, conditioned on the input, such deep networks are equivalent to an affine transformation, but globally, the transformation is not convex.

**Smoothing nonlinearity by smoothing the region assigning process.**   For completeness, we note that Eq. (3) can be written in close form as:

$$\mathbf{t}_r^\beta = \frac{\exp\left(\frac{\beta(\langle \mathbf{A}_{r,\cdot},\mathbf{x}\rangle + \mathbf{b}_r)}{1-\beta}\right)}{\sum_{i=1}^R \exp\left(\frac{\beta(\langle \mathbf{A}_{i,\cdot},\mathbf{x}\rangle + \mathbf{b}_i)}{1-\beta}\right)} \quad \text{for } r = 1, \ldots, R. \tag{7}$$

Using Eq. (3) and a ReLU activation function, switching from $\beta = 1$ to $\beta = 0.5$ is provably equivalent to replacing ReLU with the Sigmoid Linear Unit (SiLU). In the limit as $\beta \to 0$, the activation function becomes linear—thus making the entire input-output mapping of the network linear as well.

**Smoothing nonlinearity by smoothing the max operation**   Instead of relying on a soft region assignment, we can instead directly smooth the maximum function. It is already well known that smoothing the maximum operator leads to the log-sum-exp operator (i.e. SoftPlus). Hence, the mapping from Eq. (1) in close form becomes

$$(1-\beta)\ln\left[\sum_{r=1}^R \exp\left(\frac{\langle \mathbf{A}_{r,\cdot},\mathbf{x}\rangle + \mathbf{b}_r}{1-\beta}\right)\right], \tag{8}$$

where we parameterized the mapping so that its behavior is akin to Eq. (3), a value of $\beta \to 1$ recovers the original affine spline activation, e.g., ReLU.

**CTU: Combining smoothing strategies**   Building on the MASO interpretation, curvature tuning proposes to smoothen non-linearities (e.g. ReLU) of a DN as a novel form of model steering, that avoids retraining or fine-tuning the learned layers. By recalling Section 3, when smoothing is performed by applying Eq. (3) or Eq. (4) to a DN layer (interpreted as a MASO), the layer's output is statistically biased by either a negative or a positive factor, respectively. In order to counter the bias without retraining, a convex combination of the two equations is used, as shown in Fig. 2 for different values of $\beta$.

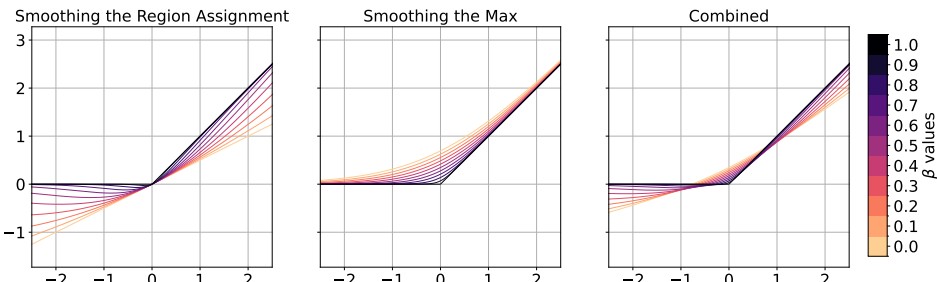

Figure 2: Visualization of non-linearity smoothing through region assignment smoothing, max smoothing, and their combination. **The combined approach mitigates the opposing biases introduced by the individual methods.**

# B  Supplementary experimental details

This section provides additional experimental setup details and results, organized to correspond with the subsections in Section 4.

All experiments were conducted using 8 RTX 3090 GPUs and one L40 GPU, with runs performed under random seeds 42, 43, and 44.

## B.1  Improving generalization on downstream datasets with *CT*

The downstream datasets we use include Arabic Characters [18], Arabic Digits [19], Beans [20], CUB-200-2011 [21], DTD [22], FashionMNIST [23], FGVC-Aircraft [24], Flowers102 [25], Food101 [26], and three subsets from MedMNIST-PathMNIST, OCTMNIST, and DermaMNIST [27]. For each of the 12 downstream datasets, we split the data into training, validation, and test sets. If a dataset does not include a validation set, we hold out 20% of the training data using stratified sampling. Otherwise, we use the original validation split provided.

To apply *CT*, we replace all ReLUs in the backbone with CTUs, freeze the backbone weights, and train a new linear classifier on the penultimate layer. The optimal $\beta$ is selected via grid search over $\beta \in [0.7, 1]$ with a step size of 0.01. The linear classifiers are trained for 20 epochs using the Adam optimizer with a learning rate of $10^{-3}$, employing linear warm-up during the first epoch and decaying the learning rate by a factor of 10 after epoch 10. The linear probing baseline follows the same training configuration.

For both *CT* and linear probing, models are trained on the training split of each downstream dataset, with the checkpoint achieving the highest validation accuracy selected for evaluation on the test set.

Additional results are provided as follows:

- Table 3: mean accuracy over three runs of ImageNet-pretrained ResNet-152 when transferred to 12 downstream datasets, comparing linear probing with and without *CT*.

- Table 4: average optimal $\beta$ values for *CT* across three runs.

- Fig. 3: example validation accuracy vs. $\beta$ curves over three runs for *CT*.

As shown in Table 4, the average optimal $\beta$ values for *CT* across datasets are 0.84 for ResNet-18, 0.94 for ResNet-50, and 0.96 for ResNet-152. These values are consistently close to 1, suggesting the search range can be narrowed for efficiency. The upward trend with model size indicates that larger models require less curvature adjustment, which is intuitive as deeper networks can approximate complex curvature more effectively. Example accuracy curves in Fig. 3 show that accuracy varies smoothly with $\beta$ and typically peaks in the middle of the search range.

Table 3: Mean accuracy (%) over three runs of ImageNet-pretrained ResNet-152 when transferred to 12 downstream datasets. The second row under each method indicates the number of trainable parameters (excluding the linear classifier). *CT* **outperforms linear probing on the frozen backbone, and** *Trainable CT* **surpasses LoRA (rank 1).**

| Dataset | Frozen (0) | *CT* (1) | LoRA (243283) | *Train CT* (143744) |
|---|---|---|---|---|
| Arabic Characters | 79.86 | 79.21 | 95.96 | **96.47** |
| Arab Digits | 98.07 | 98.15 | **99.15** | 99.10 |
| Beans | 87.50 | 87.50 | 93.75 | **96.35** |
| CUB-200 | 67.68 | 68.15 | 70.59 | **73.04** |
| DTD | 66.97 | **66.99** | 66.63 | 63.39 |
| FashionMNIST | 90.44 | 90.51 | 92.77 | **93.39** |
| FGVC-aircraft | 38.74 | 38.51 | 48.84 | **58.16** |
| Flowers102 | 82.98 | 83.28 | **84.40** | 83.43 |
| Food101 | 71.11 | 71.13 | 74.66 | **76.08** |
| DermaMNIST | 75.68 | 76.23 | 76.91 | **77.94** |
| OCTMNIST | 69.27 | 69.10 | **76.43** | 75.17 |
| PathMNIST | **89.91** | 89.82 | 84.94 | 83.60 |
| Average | 76.52 | 76.55 | 80.42 | **81.34** |

Table 4: Mean $\beta$ of *CT* over three runs of ImageNet-pretrained ResNet-18/50/152 and Imagenette-pretrained Swin-T/S when transferred to 12 downstream datasets. **The learned $\beta$ values are consistently high (ranging from 0.84 to 0.96 across models), and tend to be larger for larger models.**

| Dataset | ResNet-18 | ResNet-50 | ResNet-152 | Swin-T | Swin-S |
|---|---|---|---|---|---|
| Arabic Characters | 0.77 | 0.89 | 0.96 | 0.92 | 0.97 |
| Arabic Digits | 0.75 | 0.93 | 0.95 | 0.86 | 0.96 |
| Beans | 0.76 | 0.94 | 0.97 | 0.94 | 0.98 |
| CUB-200 | 0.91 | 0.93 | 0.94 | 0.97 | 0.87 |
| DTD | 0.88 | 0.98 | 0.98 | 0.96 | 0.95 |
| FashionMNIST | 0.92 | 0.95 | 0.96 | 0.89 | 0.98 |
| FGVC-Aircraft | 0.82 | 0.90 | 0.95 | 0.93 | 0.97 |
| Flowers102 | 0.84 | 0.96 | 0.95 | 0.99 | 0.97 |
| Food101 | 0.87 | 0.98 | 0.99 | 0.97 | 0.99 |
| DermaMNIST | 0.94 | 0.95 | 0.95 | 0.93 | 0.89 |
| OCTMNIST | 0.80 | 0.94 | 0.98 | 0.88 | 0.95 |
| PathMNIST | 0.83 | 0.96 | 0.92 | 0.90 | 0.94 |
| Average | 0.84 | 0.94 | 0.96 | 0.93 | 0.95 |

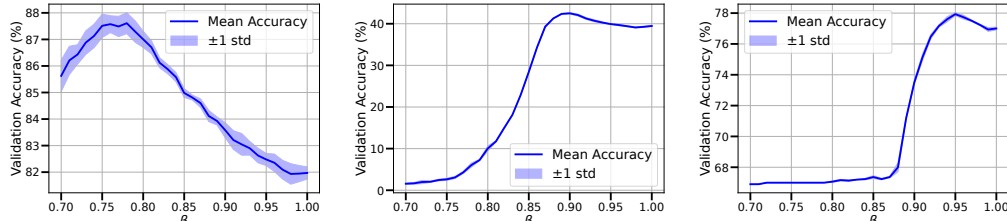

(a) ResNet-18 on Arabic Characters   (b) ResNet-50 on FGVC-Aircraft   (c) ResNet-152 on DermaMNIST

Figure 3: Validation accuracy (%) of *CT* during the $\beta$ search, averaged over three runs. **The accuracy curve varies smoothly and typically peaks in the middle of the $\beta$ range.**

## B.2  *Trainable CT* is comparable to LoRA

To apply *Trainable CT*, we replace all ReLUs in the backbone with CTUs, freeze the backbone weights, and train a new linear classifier on the penultimate layer. All $\beta$ parameters are initialized to 0.8 and all $c$ parameters to 0.5, and these are jointly trained with the linear head.

And LoRA is applied to all convolutional and linear layers in the backbone. We provide the implementation details for it in Appendix E.

Both *Trainable CT* and LoRA are trained for 20 epochs using the Adam optimizer. To ensure proper convergence, we use different learning rates: for *Trainable CT*, a learning rate of $10^{-1}$ is applied to the $(\beta, c)$ parameters and $10^{-3}$ to the linear classifier; for LoRA, a learning rate of $10^{-4}$ is used for both the adapter parameters and the classifier. As before, we apply linear warm-up during the first epoch and decay the learning rate by a factor of 10 after epoch 10. For both methods, models are trained on the training set of each downstream dataset, selected based on the highest validation accuracy, and evaluated on the test set.

Additional results are provided as follows:

- Table 3: mean accuracy over three runs of ImageNet-pretrained ResNet-152 when transferred to 12 downstream datasets, comparing LoRA and *Trainable CT*.

- Tables 5 and 6: mean and standard deviation of the learned $\beta$ and $c$ values for *Trainable CT* across three runs.

- Figs. 4 and 5: example distributions of $\beta$ and $c$ values in *Trainable CT*, illustrating commonly and uncommonly observed patterns.

Table 5: Distribution of $\beta$ values in *Trainable CT*, computed over all $\beta$ parameters across all three runs of ImageNet-pretrained ResNet-18/50/152 and Imagenette-pretrained Swin-T/S when transferred to 12 downstream datasets. **The mean and standard deviation of $\beta$ are similar across models (means between 0.69–0.77, stds between 0.31–0.37), suggesting consistent tuning behavior at the model level, while the relatively large standard deviations indicate substantial variation of $\beta$ within each network.**

| Dataset | ResNet-18 | ResNet-50 | ResNet-152 | Swin-T | Swin-S |
|---|---|---|---|---|---|
| Arabic Characters | 0.72 ± 0.34 | 0.65 ± 0.41 | 0.68 ± 0.39 | 0.73 ± 0.35 | 0.76 ± 0.33 |
| Arabic Digits | 0.70 ± 0.43 | 0.62 ± 0.48 | 0.62 ± 0.47 | 0.65 ± 0.42 | 0.64 ± 0.43 |
| Beans | 0.72 ± 0.26 | 0.76 ± 0.23 | 0.77 ± 0.19 | 0.79 ± 0.24 | 0.83 ± 0.23 |
| CUB-200 | 0.81 ± 0.17 | 0.76 ± 0.29 | 0.79 ± 0.29 | 0.82 ± 0.27 | 0.83 ± 0.28 |
| DTD | 0.78 ± 0.19 | 0.77 ± 0.25 | 0.79 ± 0.24 | 0.87 ± 0.17 | 0.88 ± 0.19 |
| FashionMNIST | 0.72 ± 0.41 | 0.65 ± 0.46 | 0.63 ± 0.46 | 0.67 ± 0.42 | 0.66 ± 0.43 |
| FGVC-Aircraft | 0.75 ± 0.23 | 0.70 ± 0.33 | 0.74 ± 0.32 | 0.81 ± 0.25 | 0.82 ± 0.27 |
| Flowers102 | 0.75 ± 0.16 | 0.75 ± 0.21 | 0.79 ± 0.17 | 0.81 ± 0.22 | 0.84 ± 0.22 |
| Food101 | 0.80 ± 0.30 | 0.71 ± 0.43 | 0.76 ± 0.40 | 0.78 ± 0.36 | 0.74 ± 0.40 |
| DermaMNIST | 0.74 ± 0.34 | 0.70 ± 0.39 | 0.70 ± 0.37 | 0.76 ± 0.32 | 0.77 ± 0.32 |
| OCTMNIST | 0.67 ± 0.45 | 0.62 ± 0.48 | 0.63 ± 0.47 | 0.76 ± 0.37 | 0.64 ± 0.45 |
| PathMNIST | 0.69 ± 0.43 | 0.65 ± 0.47 | 0.61 ± 0.48 | 0.78 ± 0.36 | 0.70 ± 0.43 |
| Average | 0.74 ± 0.31 | 0.69 ± 0.37 | 0.71 ± 0.35 | 0.77 ± 0.31 | 0.76 ± 0.33 |

For *Trainable CT*, to better understand how it behaves during training, we analyze the distributions of learned $\beta$ and $c$ values (as shown in Appendix Tables 5 and 6). We observe a high degree of within-model variation, with standard deviations ranging from 0.31 to 0.38, while the means remain remarkably stable across architectures: 0.69 to 0.74 for $\beta$ and 0.57 to 0.59 for $c$. These mean values are close to those used in *CT*, though the learned $\beta$ values tend to be smaller than the optimal shared $\beta$ found in *CT* (0.84 to 0.96), while the learned $c$ values are larger than the fixed $c = 0.5$.

We further visualize the distributions of the learned $\beta$ and $c$ values of *Trainable CT* in Appendix Figs. 4 and 5. In most datasets, as shown in Appendix Fig. 4 (OCTMNIST), both $\beta$ and $c$ exhibit a sharp U-shaped distribution—concentrating near 0 and 1 with a flat middle. This suggests that *Trainable CT* leverages its parameter flexibility to assign values at the extremes, producing an effective

Table 6: Distribution of $c$ values in *Trainable CT*, computed over all $c$ parameters across all three runs of ImageNet-pretrained ResNet-18/50/152 and Imagenette-pretrained Swin-T/S when transferred to 12 downstream datasets. **The three ResNet models exhibit similar distributions (means around 0.57–0.59, stds around 0.36–0.38), while the two Swin models also show comparable statistics with higher means (0.67–0.70), and similar stds (0.35-0.37). All models display substantial within-network variation, and the higher average $c$ in Swin models may reflect insufficient pretraining relative to the ResNets.**

| Dataset | ResNet-18 | ResNet-50 | ResNet-152 | Swin-T | Swin-S |
|---|---|---|---|---|---|
| Arabic Characters | 0.63 ± 0.39 | 0.61 ± 0.39 | 0.57 ± 0.37 | 0.66 ± 0.41 | 0.70 ± 0.38 |
| Arabic Digits | 0.59 ± 0.43 | 0.57 ± 0.42 | 0.55 ± 0.41 | 0.63 ± 0.45 | 0.71 ± 0.43 |
| Beans | 0.61 ± 0.29 | 0.54 ± 0.25 | 0.53 ± 0.23 | 0.67 ± 0.26 | 0.69 ± 0.24 |
| CUB-200 | 0.60 ± 0.37 | 0.63 ± 0.37 | 0.60 ± 0.34 | 0.70 ± 0.33 | 0.70 ± 0.33 |
| DTD | 0.59 ± 0.31 | 0.60 ± 0.32 | 0.57 ± 0.30 | 0.68 ± 0.25 | 0.74 ± 0.24 |
| FashionMNIST | 0.55 ± 0.44 | 0.60 ± 0.42 | 0.56 ± 0.42 | 0.62 ± 0.46 | 0.69 ± 0.43 |
| FGVC-Aircraft | 0.61 ± 0.36 | 0.63 ± 0.37 | 0.58 ± 0.35 | 0.71 ± 0.33 | 0.68 ± 0.33 |
| Flowers102 | 0.58 ± 0.26 | 0.54 ± 0.26 | 0.54 ± 0.23 | 0.65 ± 0.29 | 0.66 ± 0.25 |
| Food101 | 0.46 ± 0.47 | 0.63 ± 0.44 | 0.60 ± 0.43 | 0.72 ± 0.42 | 0.76 ± 0.39 |
| DermaMNIST | 0.58 ± 0.38 | 0.59 ± 0.37 | 0.57 ± 0.36 | 0.66 ± 0.36 | 0.71 ± 0.33 |
| OCTMNIST | 0.55 ± 0.45 | 0.60 ± 0.42 | 0.57 ± 0.42 | 0.58 ± 0.47 | 0.65 ± 0.45 |
| PathMNIST | 0.51 ± 0.45 | 0.58 ± 0.43 | 0.57 ± 0.42 | 0.71 ± 0.42 | 0.76 ± 0.40 |
| Average | 0.57 ± 0.38 | 0.59 ± 0.37 | 0.57 ± 0.36 | 0.67 ± 0.37 | 0.70 ± 0.35 |

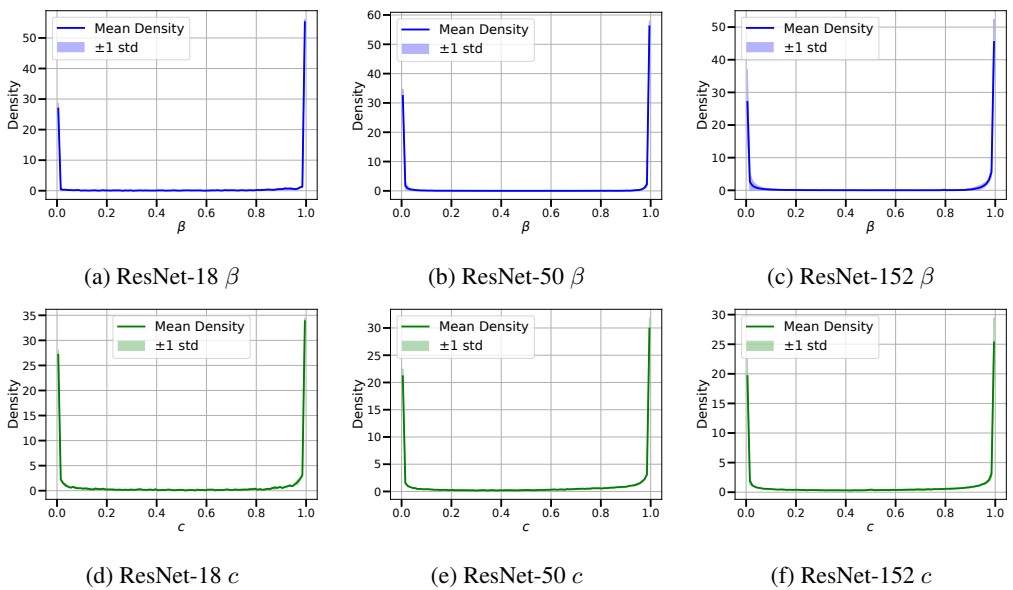

(a) ResNet-18 $\beta$     (b) ResNet-50 $\beta$     (c) ResNet-152 $\beta$

(d) ResNet-18 $c$     (e) ResNet-50 $c$     (f) ResNet-152 $c$

Figure 4: Common distributions of $\beta$ (top) and $c$ (bottom) in *Trainable CT* across ResNet-18/50/152, averaged over three runs (OCTMNIST shown as a representative dataset). **Both $\beta$ and $c$ consistently exhibit sharp U-shaped distributions that appear similar across all models.**

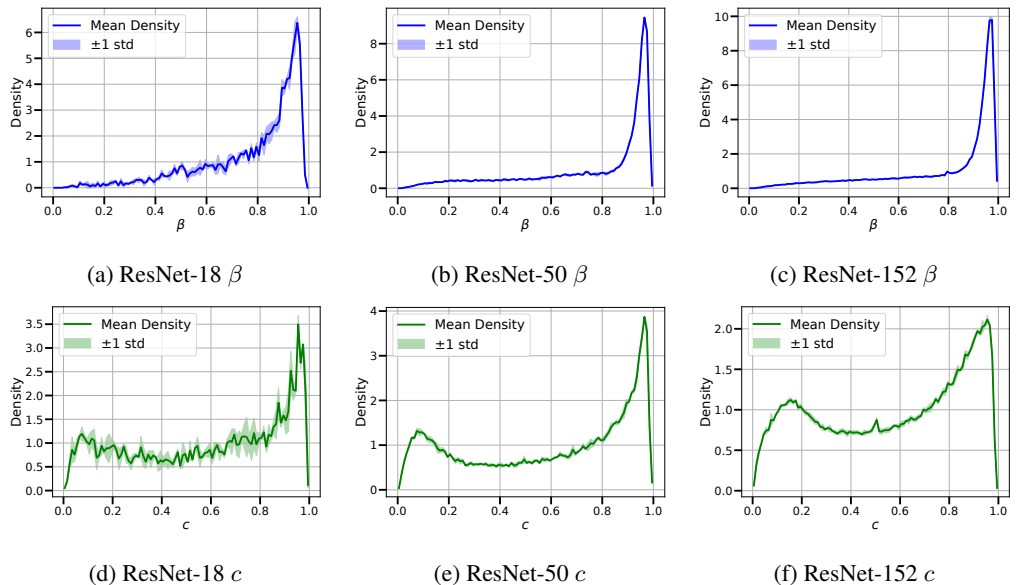

Figure 5: Uncommon distributions of $\beta$ (top) and $c$ (bottom) in *Trainable CT* across ResNet-18/50/152, averaged over three runs (DTD shown as an example dataset). **While the overall shape is dataset-specific, the distributions of both $\beta$ and $c$ remain consistent across models.**

average close to the manually chosen settings in *CT*, rather than concentrating around the mean values themselves.[2] In a few datasets, we observe deviations from this trend, as exemplified in Appendix Fig. 5 (DTD). Nonetheless, a consistent pattern is that for any given dataset, the distributions remain visually similar across all models.

## B.3  CT shows promise on transformers and emerging architectures

In this subsection, we investigate the effectiveness of *Trainable CT* on transformer architectures. Unlike ResNets, transformers incorporate attention layers and typically use non-piecewise-affine activation functions (e.g., SiLU, GELU), which fall outside the max-affine spline framework and thus weaken theoretical guarantees. Nevertheless, we validate their effectiveness empirically.

We consider Swin-T/S [28], whose activation function is GELU. Since GELU can be closely approximated by a CTU with $\beta = 0.6403$ and $c = 1$, we initialize all CTU parameters in *Trainable CT* with these values. We compare against both the linear probing baseline and LoRA (rank $r = 1$, scale $\alpha = 1$), where LoRA is applied to all QKV projection layers.

For fairness, we cross-validate the learning rate for each method. Specifically, for *Trainable CT*, we test initial learning rates of $10^{-1}$, $10^{-2}$, and $10^{-3}$ for the $(\beta, c)$ parameters. For the linear probing baseline, we use $10^{-2}$, $10^{-3}$, and $10^{-4}$, and for LoRA, $10^{-3}$, $10^{-4}$, and $10^{-5}$. We report the best performance achieved for each method across these choices.

The results in Table 7 show that *Trainable CT* yields average relative improvements of 0.61% and 1.76% over the frozen baseline on Swin-T and Swin-S, respectively, but trails LoRA by 3.45% and 4.61%. Notably, *Trainable CT* achieves this competitive performance with only 0.71% and 0.58% as many trainable parameters as LoRA on Swin-T and Swin-S, a much smaller ratio than in the ResNet experiments. Importantly, since *Trainable CT* operates orthogonally to other PEFT methods such as LoRA, the goal is not to demonstrate that CT surpasses them, but rather that it can be combined with them to further enhance performance. Thus, even though *Trainable CT* underperforms LoRA in this setting, the results highlight its potential on transformer models.

---

[2]This behavior may in part be influenced by the sigmoid-based parameterization used in our implementation of *Trainable CT* to constrain $\beta$ and $c$ during training.

Table 7: Accuracy (%) of Imagenet-pretrained Swin-T/S when transferred to 12 downstream datasets. The second row under each method indicates the number of trainable parameters (excluding the linear classifier). ***Trainable CT* outperforms linear probing but underperforms LoRA.**

| Dataset | Swin-T | | | Swin-S | | |
|---|---|---|---|---|---|---|
| | Frozen (0) | LoRA (74832) | *Train CT* (532) | Frozen (0) | LoRA (148560) | *Train CT* (868) |
| Arabic Characters | 83.27 | **93.57** | 86.10 | 83.78 | **94.58** | 86.76 |
| Arabic Digits | 98.24 | **99.12** | 98.39 | 98.32 | **99.18** | 98.39 |
| Beans | 89.84 | **95.31** | 92.19 | **92.97** | **92.97** | 92.19 |
| CUB-200 | 73.65 | **77.60** | 74.23 | 72.61 | **79.98** | 73.42 |
| DTD | 71.17 | 70.32 | **71.86** | 70.16 | 70.48 | **72.82** |
| FashionMNIST | 89.75 | **92.95** | 90.25 | 89.85 | **93.45** | 89.96 |
| FGVC-Aircraft | 47.61 | 47.16 | **47.73** | 44.52 | **52.42** | 45.09 |
| Flowers102 | 86.88 | **90.57** | 85.41 | 83.28 | **90.29** | 85.04 |
| Food101 | 77.05 | **83.23** | 78.97 | 77.72 | **85.50** | 79.60 |
| DermaMNIST | 76.51 | **77.11** | 75.76 | 76.66 | **77.41** | **77.41** |
| OCTMNIST | 69.10 | **76.70** | 67.30 | 67.00 | **77.00** | 69.70 |
| PathMNIST | 90.65 | **92.56** | 91.84 | 89.89 | **92.60** | 92.08 |
| **Average** | 79.48 | **83.02** | 80.00 | 78.90 | **83.82** | 80.21 |

## B.4 Improving robustness on adversarial and corrupted data with *CT*

Due to computational constraints, we evaluate each benchmark using 1,000 samples. For adversarial evaluations, we follow the official RobustBench settings: $\varepsilon_2 = 0.5$ for $\ell_2$ attacks and $\varepsilon_\infty = \frac{8}{255}$ for $\ell_\infty$ attacks.

## B.5 Improving $\ell_\infty$ robustness with *Trainable CT*

In Section 4, we showed that *CT* can significantly improve $\ell_\infty$ robustness by adjusting the curvature of the model's decision boundary, without relying on labeled data or explicit loss functions. Since *Trainable CT* also directly modulates decision boundary curvature, it is expected to yield similar effects. In this subsection, we demonstrate that *Trainable CT* can indeed improve $\ell_\infty$ robustness as a natural byproduct of standard finetuning, even without explicitly targeting adversarial robustness.

To evaluate this, we extend the RobustBench benchmark to *Trainable CT* and LoRA. Specifically, we transfer ImageNet-pretrained ResNet-18/50/152 models to CIFAR-10/100 using the same setup as in Section 4—linear probing, *Trainable CT*, and LoRA—and then assess $\ell_\infty$ robustness under attack using RobustBench.

Table 8: Mean robust accuracy (%) over three runs of ImageNet-pretrained ResNet-18/50/152 transferred to CIFAR-10/100 under $\ell_\infty$ attack. ***Trainable CT* substantially enhances $\ell_\infty$ robustness as a byproduct of finetuning, whereas LoRA provides limited or even negative gains.**

| Model | Dataset | Frozen | LoRA | **Train CT** |
|---|---|---|---|---|
| ResNet18 | CIFAR10 | 0.30 | 0.70 | **1.57** |
| | CIFAR100 | 0.03 | 0.07 | **0.17** |
| | Average | 0.17 | 0.38 | **0.87** |
| ResNet50 | CIFAR10 | 0.20 | 0.33 | **2.43** |
| | CIFAR100 | 0.00 | 0.03 | **0.07** |
| | Average | 0.10 | 0.18 | **1.25** |
| ResNet152 | CIFAR10 | 0.43 | 0.20 | **5.10** |
| | CIFAR100 | 0.17 | 0.00 | **0.00** |
| | Average | 0.30 | 0.10 | **2.55** |

As shown in Appendix Table 8, *Trainable CT* achieves average relative improvements of 411.11%, 1116.67%, and 488.46% over linear probing on ResNet-18/50/152, respectively. Furthermore, it consistently surpasses LoRA by 136.90%, 365.00%, and 2450.00% on the same architectures. These results indicate that, even without explicit adversarial training, *Trainable CT* substantially enhances $\ell_\infty$ robustness by directly modulating decision boundary curvature. LoRA, by contrast, leaves activation nonlinearities unchanged and thus offers limited or even negative robustness benefits. This empirical finding underscores the practical advantage of *Trainable CT*: by shaping decision boundary curvature, it yields direct gains in adversarial robustness without relying on adversarial objectives.

## C  Theoretical Intuition

This section provides theoretical intuition behind Curvature Tuning. Section C.1 casts CT as a projection over a space of smooth functions, while Section C.2 provides a toy example illustrating how CT can improve approximation of a target function of non-vanishing curvature, upon an ideal baseline ReLU network.

### C.1  CT Operates as a Projection

At its core, Curvature Tuning operates by modulating the non-linearity of the activation functions of a trained model, providing a novel approach to model steering. In order to formalize the effect of CT, the following briefly introduces the notion of spaces of smooth functions.

**Sobolev spaces**   Let $f : \mathbb{R}^d \to \mathbb{R}$ be a function and $\Omega \subseteq \mathbb{R}^d$ be a bounded domain. For $1 \le p < \infty$, define $L^p(\Omega)$ as the space of functions $f : \Omega \to \mathbb{R}$ such that the $L^p$ norm is finite, i.e.

$$\|f\|_{L^p(\Omega)} := \left( \int_\Omega |f(\mathbf{x})|^p d\mathbf{x} \right)^{\frac{1}{p}} < \infty \tag{9}$$

Let $\alpha = (\alpha_1, \ldots, \alpha_d)$ denote a multi-index, with $|\alpha| := \sum_i^d \alpha_i$, and $\alpha_i \in \mathbb{N}, \forall i = 1, \ldots, d$. Let $q \in \mathbb{N}^*$. For $|\alpha| > 0$, define the Sobolev semi-norm

$$|f|_{W^{q,p}(\Omega)} := \left( \sum_{|\alpha| \le q} \|D^\alpha f\|_{L^p(\Omega)}^p \right)^{\frac{1}{p}} \tag{10}$$

with $D^\alpha f := \frac{\partial^{|\alpha|} f}{\partial x_1^{\alpha_1} \ldots \partial x_d^{\alpha_d}}$ denoting $|\alpha|$-th order partial derivatives of $f$. Define the Sobolev norm

$$\|f\|_{W^{q,p}(\Omega)} := \left( \|f\|_{L^p(\Omega)}^p + |f|_{W^{q,p}(\Omega)}^p \right)^{\frac{1}{p}} \tag{11}$$

and the Sobolev space $W^{q,p}(\Omega) := \{f : \Omega \to \mathbb{R} \text{ s.t. } \|f\|_{L^p(\Omega)}^p + |f|_{W^{q,p}(\Omega)}^p < \infty\}$.

For a finite set $\mathcal{D} = \{\mathbf{x}_i\}_{i=1}^n$, the Sobolev semi-norm becomes

$$|f|_{W^{q,p}(\mathcal{D})} := \left( \sum_{|\alpha| \le q} \frac{1}{n} \sum_{i=1}^n \|D^\alpha f(\mathbf{x}_i)\|_p^p \right)^{\frac{1}{p}} \tag{12}$$

Finally, for $\mathbf{x} \in \mathbb{R}^d$, let $\|\mathbf{x}\|_p$ denote the $p$-norm, corresponding to the Euclidean norm for $p = 2$.

**Curvature Tuning acts as a Sobolev Projection**   To characterize Curvature Tuning, we are interested in the space $W^{2,2}(\Omega)$, equipped with the Sobolev semi-norm

$$|f|_{W^{2,2}(\Omega)}^2 = \|\nabla_{\mathbf{x}} f\|_{L_2(\Omega)}^2 + \|\nabla_{\mathbf{x}}^2 f\|_{L_2(\Omega)}^2 \tag{13}$$

We begin by considering the Sobolev semi-norm of a ReLU network (equivalent to the case of Eq. (5) with $\beta \to 1$). For each $\mathbf{x} \in \mathbb{R}^d$, the gradient of a ReLU network

$$f(\mathbf{x}) = \left( W^L \circ \varphi \circ \ldots \circ \varphi \circ W^1 \right) (\mathbf{x}) \tag{14}$$

with $\varphi(z) := \max(0, z)$, for $z \in \mathbb{R}$, is given by

$$\nabla_{\mathbf{x}} f(\mathbf{x}) = W^L \prod_{\ell=L-1}^{1} D^\ell(\mathbf{x}) W^\ell \tag{15}$$

where $D^\ell(\mathbf{x})$ is a diagonal matrix with $D_{ii}^\ell(\mathbf{x}) = \mathbf{1}_{\{\mathbf{z}_i^\ell > 0\}}$, with $\mathbf{z}_i^\ell = W_i^\ell \mathbf{z}^{\ell-1} + \mathbf{b}_i^\ell$ denoting the pre-activation of the $\ell$-th layer, for $\ell = 1, \dots, L$, with $\mathbf{z}^0 := \mathbf{x}$.

We make the following observations:

O1 Since ReLU networks are differentiable a. e., the gradients $\nabla_{\mathbf{x}} f(\mathbf{x})$ are bounded in norm by the network's Lipschitz constant, which can be defined as $C = \sup_{\mathbf{x} \in \Omega} \|\nabla_{\mathbf{x}} f(\mathbf{x})\|_2$. Hence, for $\Omega = \mathcal{D}$, the Lipschitz constant provides an upper bound on the first-order term of the Sobolev semi-norm in Equation 13.

O2 Finally, we observe that since ReLU networks express piece-wise affine functions, the Hessian norm vanishes a.e. (i.e. wherever the Hessian is well defined), providing a bound on the second-order term of Equation 13.

Equipped with the above observations, in the following we characterize CT.

**Theorem C.1.** *Let $f : \mathbb{R}^d \to \mathbb{R}$ denote a ReLU network, with model parameter $\mathbf{W}$ collecting all weights and biases. For $c \in [0, 1]$ and fixed $\beta \in [0, 1)$, replacing every instance of ReLU with a CTU (Equation 5) with hyperparameters $\beta, c$ is equivalent to projecting $f$ to a smooth function $f_{\beta,c} \in W^{2,2}(\Omega)$ in the Sobolev space $W^{2,2}(\Omega)$, with bounded Sobolev semi-norm.*

*Particularly, it holds $\|\nabla_{\mathbf{x}}^2 f(\mathbf{x})\|_{L^2(\Omega)} \le \|\nabla_{\mathbf{x}}^2 f_{\beta,c}(\mathbf{x})\|_{L^2(\Omega)}$, from which $f_{\beta,a}$ enjoys higher local expressivity (non-vanishing curvature), while retaining the same model parameter $\mathbf{W}$.*

Before proving Theorem C.1, we state the following Lemma, bounding the derivative of a CTU.

**Lemma C.2.** *Let $\varphi_{\beta,c}(x)$ be defined according to Eq. (5), for $\beta \in [0, 1)$ and $c \in [0, 1]$. Then*

$$\varphi_{\beta,c}'(x) = c\left(\sigma(bx) + bx\sigma(bx)(1 - \sigma(bx))\right) + (1 - c)\sigma\left(\frac{bx}{\beta}\right) \tag{16}$$

*where $b := \frac{\beta}{1-\beta}$ and $\sigma(x) = \frac{\exp x}{1+\exp x}$ is the sigmoid activation.*

*Furthermore, $\exists \, \overline{h}_b \in \mathbb{R}^+$ such that*

$$-c\overline{h}_b \le \varphi_{\beta,c}'(x) \le 1 + c\overline{h}_b \qquad \forall x \in \mathbb{R}, \quad \beta \in [0, 1) \tag{17}$$

*Proof.* We recall that, since $\forall x \in \mathbb{R}$, $\varphi_{\beta,c}(x)$ is defined as the convex combination of the SiLU activation function ($c = 1$) and the SoftPlus activation ($c = 0$), we can bound $\varphi_{\beta,c}'(x)$ by the convex combination of individual bounds obtained for the cases $c = 0$ and $c = 1$.

**SoftPlus**. If $c = 0$, then $\varphi_{\beta,0}'(x) = \sigma\left(\frac{x}{1-\beta}\right)$ and $0 \le \varphi_{\beta,0}'(x) \le 1 \, \forall x$, since the derivative is defined as a sigmoid.

**SiLU**. If $c = 1$, $\varphi_{\beta,1}'(x) = \sigma(bx) + bx\sigma(bx)(1 - \sigma(bx))$. The first term in the sum is bounded by definition of sigmoid. For the second term, we note that $\sigma(bx)(1 - \sigma(bx))$ is also bounded, and achieves it maximum at $x = 0$, for which $0 \le \sigma(bx)(1 - \sigma(bx)) \le \frac{1}{4}$. Furthermore, in the limit $x \to +\infty$, it holds $\varphi_{\beta,1}'(x) \to 1$, while $\varphi_{\beta,1}'(x) \to 0$ for $x \to -\infty$.

In the non-asympototic regime, $\sigma(bx)(1 - \sigma(bx)) > 0$, and so the maximum value of $bx\sigma(bx)(1 - \sigma(bx))$ also depends on $bx$. To bound $\varphi_{\beta,c}'$ in this case, let us first consider $x > 0$. By defining $\overline{h}_b = \max_{bx \ge 0} bx\sigma(bx)(1 - \sigma(bx))$, then we finally obtain $0 \le \varphi_{\beta,1}'(x) \le 1 + \overline{h}_b$.

For the case $x < 0$, by using the identity $\sigma(x) = 1 - \sigma(-x)$, we have that $-\overline{h}_b \le \varphi_{\beta,1}'(x) \le 1$. By combining the results, we have

$$-\overline{h}_b \le \varphi_{\beta,1}'(x) \le 1 + \overline{h}_b \qquad \forall x \in \mathbb{R}, \quad \beta \in [0, 1) \tag{18}$$

In conclusion, by convex combination of cases $c = 0$ and $c = 1$, Eq. (18) holds uniformly in $x$. $\quad\square$

We can now prove Theorem C.1. To do so, for $f_{\beta,c}$ we have to show that

1. $f_{\beta,c}$ is smooth in $\mathbf{x}$, for $\mathbf{x} \in \Omega$
2. $\|f_{\beta,c}\|_{W^{2,2}(\Omega)} < \infty$

for a network $f_{\beta,c}$ obtained by replacing every ReLU $\varphi$ with a CTU $\varphi_{\beta,c}$, while keeping all learned parameters $\mathbf{W}$ fixed.

*Proof.* We provide a proof for $\Omega = \mathcal{D} = \{\mathbf{x}_i\}_{i=1}^n$, under the common i.i.d. assumption on $\mathcal{D}$.

To prove the first point, we observe that for $\beta \in [0,1)$, the CTU activation function is smooth, i.e. $\varphi_{\beta,c} \in \mathcal{C}^\infty(\mathbb{R})$, thus making the whole network $f_{\beta,c}$ smooth.

We now consider the Sobolev semi-norm $|f_{\beta,c}|_{W^{2,2}(\Omega)}$. Starting with the first-order gradient, by recalling that CT replaces each occurrence of ReLU with the CTU activation function (Equation 5), the input gradient of CT is given by

$$\nabla_{\mathbf{x}} f_{\beta,c}(\mathbf{x}) = W^L \prod_{\ell=L-1}^1 D_{\beta,c}^\ell(\mathbf{z}^\ell) W^\ell \tag{19}$$

where $D_{\beta,c}^\ell(\mathbf{z}^\ell) = \mathrm{diag}(\varphi'_{\beta,c}(\mathbf{z}^\ell))$ with $\varphi'_{\beta,c}(\mathbf{z}^\ell)_i := \varphi'_{\beta,c}(\mathbf{z}_i^\ell)$ according to Eq. (16).

To bound the Jacobian norm, we observe that

$$\|\nabla_{\mathbf{x}} f_{\beta,c}(\mathbf{x})\| = \|W^L \prod_{\ell=L-1}^1 D_{\beta,c}^\ell(\mathbf{z}^\ell) W^\ell\| \tag{20}$$

$$\leq \|W^L\| \prod_{\ell=L-1}^1 \|D_{\beta,c}^\ell(\mathbf{z}^\ell)\| \|W^\ell\| \tag{21}$$

$$\leq \|W^L\| \prod_{\ell=L-1}^1 \sqrt{d_\ell}(1 + c\overline{h}_b) \|W^\ell\| < \infty \qquad \text{(Lemma C.2)} \tag{22}$$

independent of $\mathbf{x}$, for $W^\ell \in \mathbb{R}^{d_\ell \times d_{\ell-1}}$, with $d_0 := d$.

We now bound the second order term. By recalling that, for every $\mathbf{x} \in \mathbb{R}^d$, the Hessian $\mathbf{H}(\mathbf{x}) = \nabla_{\mathbf{x}}^2 f_{\beta,c}(\mathbf{x})$ is symmetric positive-definite, then for $\Omega = \mathcal{D}$ it holds

$$\|\nabla_{\mathbf{x}}^2 f_{\beta,c}\|_{L_2(\mathcal{D})}^2 = \frac{1}{n} \sum_{i=1}^n \|\mathbf{H}(\mathbf{x}_i)\|_2^2 \leq \max_{1 \leq i \leq n} \lambda_{\max}^2(\mathbf{H}(\mathbf{x}_i)) d_\ell < \infty \tag{23}$$

with $\lambda_{\max}(\mathbf{H}(\mathbf{x}_i))$ denoting the largest singular value of $\mathbf{H}(\mathbf{x}_i)$.

Importantly, since a ReLU network $f$ has vanishing curvature a.e., then for $0 \leq \beta < 1$, we have
$$\|\nabla_{\mathbf{x}}^2 f(\mathbf{x})\| \leq \|\nabla_{\mathbf{x}}^2 f_{\beta,c}(\mathbf{x})\|.$$

Lastly, we note that, whenever $\Omega$ is a finite discrete set $\mathcal{D}$, $f_{\beta,c}$ is measurable, ensuring that $\|f_{\beta,c}\|_{W^{2,2}(\Omega)} < \infty$, concluding the proof. □

Theorem C.1 shows that CT operates by projecting a ReLU network $f$ to a smooth function $f_{\beta,c}$ in a restricted Sobolev space. Crucially, $f_{\beta,c}$ enjoys bounded gradients (and so is well behaved), and non-vanishing local-curvature for $0 < \beta < 1$, making it locally more expressive than the affine spline $f$, for fixed $\mathbf{W}$.

Furthermore, for fixed $(\beta, c)$, CT indeed operates as a projection, since replacing every ReLU with $\varphi_{\beta,c}$ is idempotent. Importantly, while for the original ReLU network $f \in W^{2,2}(\Omega)$ the derivatives $D^\alpha f$ are understood in a weak-sense, for $c \in [0,1]$ and $\beta \in [0,1)$, $f_{\beta,c}$ belongs to a Sobolev space $W_{\mathrm{str}}^{2,2}(\Omega) \subset W^{2,2}(\Omega)$ of smooth functions, whereby the derivative $D^\alpha f_{\beta,c}$ are understood in the strong (i.e. classical) sense.

We leave for future work extending our result to *Train CT*, which is associated with a non-convex optimization problem of finding optimal $(\beta, c)$ for every neuron in the network. An additional important direction is to more closely compare $\|\nabla_{\mathbf{x}} f\|$ and $\|\nabla_{\mathbf{x}} f_{\beta,c}\|$, which may reveal more precise Lipschitz behaviour for CT, potentially better guiding the search for $\beta$ and $c$.

**CT provably controls decision boundary curvature**  To conclude this section, we observe how varying $\beta$ modulates the curvature of the whole model function $f$ and, in turn, of the model's decision boundaries. We begin by noting that for a deep network $f : \mathbb{R}^d \to \mathbb{R}^k$, the decision boundary between any class $i$ and $j$ is given by $\{\mathbf{x} \in \mathbb{R}^d : g(\mathbf{x}) := f_i(\mathbf{x}) - f_j(\mathbf{x}) = 0\}$, for any $i, j = 1, \ldots, k$ with $i \neq j$. Particularly, $g$ is itself a deep network, sharing the same parameters as $f$ up until the penultimate layer, after which the parameter is the vector $W_i^L - W_j^L$ and bias $\mathbf{b}_i^L - \mathbf{b}_j^L$. Importantly, *when varying $\beta$ while keeping all model parameters fixed*, the Jacobian $\nabla_{\mathbf{x}} g(\mathbf{x})$ and the Hessian $\nabla_{\mathbf{x}}^2 g(\mathbf{x})$ are respectively given by the gradients and Hessian of $\mathbf{z}^{L-1}(\mathbf{x})$ – corresponding to the post-activation output of the $L-1$-th layer – weighted by $W_i^L - W_j^L$. Hence, modulating the non-linearity of activation functions via $\beta$ directly controls the curvature of both model function and its decision boundaries. [3]

Particularly, for $c = 1$ (Eq. (5)), as $\beta \to 0$, the activation becomes linear. Since modern DNs (e.g. MLP, CNN, RNN) are composed of activation functions interleaved with affine layers, it follows directly that the entire input-output mapping becomes affine when $\beta \to 0$. In this setting, the curvature of the mapping—defined as the norm of its Hessian—becomes zero. As a result, transitioning from the original DN mapping ($\beta = 1$) to the linear setting effectively modulates the network decision boundary curvature, reducing it continuously to zero in the limit. For $c < 1$, the model retains non-vanishing local curvature, while the mapping becomes smooth.

## C.2   Toy Example

We conclude the discussion by providing the full derivation for the motivating example in Section 3.

Consider a binary classification problem in $\mathbb{R}^2$, whereby one is given two classes $\{\mathbf{x} \in \mathbb{R}^2 : \|\mathbf{x}\|_2 \leq \frac{1}{2}\}$ and $\{\mathbf{x} \in \mathbb{R}^2 : \frac{3}{2} \leq \|\mathbf{x}\|_2 \leq 2\}$. The decision boundary maximizing the margin between the two classes is given by $S^1 = \{\mathbf{x} \in \mathbb{R}^2 : \|\mathbf{x}\| = 1\}$.

For a ReLU network $f : \mathbb{R}^2 \to \mathbb{R}$, the maximum margin boundary is recovered by assigning $f(\mathbf{x}) = 0 \; \forall \mathbf{x} \in S^1$, for which $\sigma(f(\mathbf{x})) = 0.5$. To measure the approximation error $e$, the boundary is parameterized by $\boldsymbol{\gamma}(t) = (\cos 2\pi t, \sin 2\pi t)$, for $t \in [0, 1]$.

Then, the error is expressed by the line integral $e = \int_{\gamma} |f| d\mathbf{x} = \int_0^1 |f(\boldsymbol{\gamma}(t))| \|\boldsymbol{\gamma}'(t)\| dt$. Since $f$ expresses an Affine Spline Operator, and each linear region in $\Omega$ is convex, then the integral along $\gamma$ can be broken down into the integral along the intersection of $\gamma$ with the spline partition $\Omega$, i.e. $\Omega_{\gamma} := \Omega \cap S^1$. Importantly, this allows to pull back the affine spline breakpoints from $\Omega_{\gamma}$ to $[0, 1]$, so that $0 \leq t_1 \leq \ldots \leq t_{r'} = 1$, where $r' = |\Omega_{\gamma}|$. Then,

$$e = \int_0^1 |f(\boldsymbol{\gamma}(t))| \|\boldsymbol{\gamma}'(t)\| dt \tag{24}$$

$$= 2\pi \sum_{k=1}^{r'-1} \int_{t_k}^{t_{k+1}} |\mathbf{A}_{k,\cdot} \boldsymbol{\gamma}(t) + \mathbf{b}_k| dt \tag{25}$$

$$= 2\pi \sum_{k=1}^{r'-1} \int_{t_k}^{t_{k+1}} (-1)^z \left( \mathbf{A}_{k,\cdot} \boldsymbol{\gamma}(t) + \mathbf{b}_k \right) dt \tag{26}$$

with $z := \mathbf{1}_{\{\mathbf{A}_{k,\cdot} \boldsymbol{\gamma}(t) + \mathbf{b}_k < 0\}}$. Then,

$$e = 2\pi \sum_{k=1}^{r'-1} \int_{t_k}^{t_{k+1}} (-1)^z \left( \mathbf{A}_{k,1} \cos 2\pi t + \mathbf{A}_{k,2} \sin 2\pi t + \mathbf{b}_k \right) dt \tag{27}$$

$$= 2\pi \sum_{k=1}^{r'-1} \int_{t_k}^{t_{k+1}} (-1)^z \left( \mathbf{A}_{k,1} \frac{\sin 2\pi}{2\pi} - \mathbf{A}_{k,2} \frac{\cos 2\pi}{2\pi} + \mathbf{b}_k t \right) \Big|_{t_k}^{t_{k+1}} \tag{28}$$

$$\tag{29}$$

---

[3] In this paper, unless specified, we will thus refer interchangeably to the curvature of a DN mapping and that of its decision boundaries, whenever modulating non-linearities via CT.

which evaluates to

$$e = \sum_{k=1}^{r'-1} (-1)^z \left( 2\pi \mathbf{b}_k (t_{k+1} - t_k) + \right.$$
$$\left. + \mathbf{A}_{k,1} \left( 2 \sin \frac{t_{k+1} - t_k}{2} \cos \frac{t_{k+1} - t_k}{2} \right) - \mathbf{A}_{k,2} \left( 2 \sin \frac{t_{k+1} + t_k}{2} \sin \frac{t_k - t_{k+1}}{2} \right) \right)$$

(30)

from which clearly $e \to 0 \iff t_{k+1} \to t_k \quad \forall k$.

Hence, assuming the ReLU network considered attained optimal approximation error $e > 0$, reducing the error further requires increasing the number of breakpoints of the ASO, in turn requiring a degree of retraining (either through PEFT or training from scratch). With this view, Curvature Tuning opens an additional avenue for model adaptation: steering the model's decision boundaries by modulating the non-linearity of the activation function, allowing to tune a model towards optimality without expensive retraining. To this end, it is important to note that modulating decision boundaries is orthogonal to feature adaptation and finetuning, since it allows to change the shape of decision boundaries while keeping the model parameter $\mathbf{W}$ fixed.

# D   Curvature Tuning (CT) implementation

The following code provides the Python implementation for *CT* and *Trainable CT*:

- `CTU` & `TrainableCTU`: classes that define the CTU module used in *CT* and *Trainable CT*, respectively.
- `replace_module` & `replace_module_dynamic`: functions that apply the appropriate module replacement to integrate *CT* or *Trainable CT* into a model.

```python
import torch
from torch import nn
import torch.nn.functional as F

class CTU(nn.Module):
    """
    CTU for CT.
    """
    def __init__(self, shared_raw_beta, shared_raw_coeff, threshold
    =20):
        super().__init__()
        self.threshold = threshold
        self._raw_beta = shared_raw_beta
        self._raw_coeff = shared_raw_coeff
        self._raw_beta.requires_grad = False
        self._raw_coeff.requires_grad = False

    @property
    def beta(self):
        return torch.sigmoid(self._raw_beta)

    @property
    def coeff(self):
        return torch.sigmoid(self._raw_coeff)

    def forward(self, x):
        beta = torch.sigmoid(self._raw_beta)
        coeff = torch.sigmoid(self._raw_coeff)
        one_minus_beta = 1 - beta + 1e-6
        x_scaled = x / one_minus_beta

        return (coeff * torch.sigmoid(beta * x_scaled) * x +
                (1 - coeff) * F.softplus(x_scaled, threshold=self.
                threshold) * one_minus_beta)
```

```python
class TrainableCTU(nn.Module):
    """
    CTU for Trainable CT.
    """
    def __init__(self, num_input_dims, out_channels, raw_beta=1.386,
    raw_coeff=0.0, threshold=20):
        super().__init__()
        self.threshold = threshold

        # Decide channel dim based on input shape
        if num_input_dims == 2 or num_input_dims == 3:  # (B, C) or (B
        , L, D)
            channel_dim = -1
        elif num_input_dims == 4: # (B, C, H, W)
            channel_dim = 1
        else:
            raise NotImplementedError(f"Unsupported input dimension {
            num_input_dims}")

        param_shape = [1] * num_input_dims
        param_shape[channel_dim] = out_channels

        # Init beta
        self._raw_beta = nn.Parameter(torch.full(param_shape, float(
        raw_beta)))

        # Init coeff
        self._raw_coeff = nn.Parameter(torch.full(param_shape, float(
        raw_coeff)))

    @property
    def beta(self):
        return torch.sigmoid(self._raw_beta)

    @property
    def coeff(self):
        return torch.sigmoid(self._raw_coeff)

    def forward(self, x):
        beta = torch.sigmoid(self._raw_beta)
        coeff = torch.sigmoid(self._raw_coeff)
        one_minus_beta = 1 - beta + 1e-63
        x_scaled = x / one_minus_beta

        return (coeff * torch.sigmoid(beta * x_scaled) * x +
                (1 - coeff) * F.softplus(x_scaled, threshold=self.
                threshold) * one_minus_beta)
```

```python
def replace_module(model, old_module=nn.ReLU, new_module=CTU, **kwargs
):
    """
    Replace all instances of old_module in the model with new_module.
    """
    device = next(model.parameters(), torch.tensor([])).device  #
    Handle models with no parameters

    # Replace modules
    for name, module in model.named_modules():
        if isinstance(module, old_module):
            ct = new_module(**kwargs).to(device)

            # Replace module in the model
            names = name.split(".")
            parent = model
            for n in names[:-1]:
                if n.isdigit():
                    parent = parent[int(n)]  # for Sequential/
                    ModuleList
                else:
                    parent = getattr(parent, n)

            last_name = names[-1]
            if last_name.isdigit():
                parent[int(last_name)] = ct  # for Sequential/
                ModuleList
            else:
                setattr(parent, last_name, ct)

    return model
```

```python
def replace_module_dynamic(model, input_shape, old_module=nn.ReLU,
new_module=TrainableCTU, **kwargs):
    """
    Replace all instances of old_module in the model with new_module
    that's dynamically created based on the number of output channels.
    """
    device = next(model.parameters(), torch.tensor([])).device
    dummy_input = torch.randn(*input_shape).to(device)

    module_metadata = {}  # name -> (num_input_dims, out_channels)
    hooks = []

    def make_hook(name):
        def hook(module, input, output):
            num_input_dims = input[0].dim()
            if num_input_dims in (2, 3):     # (B, C) or (B, L, D)
                out_channels = output.shape[-1]
            elif num_input_dims == 4:        # (B, C, H, W)
                out_channels = output.shape[1]
            else:
                raise NotImplementedError(f"Unsupported output shape {
                output.shape} in {name}")
            module_metadata[name] = (num_input_dims, out_channels)

        return hook

    # Register hooks to all modules of the target type
    for name, module in model.named_modules():
        if isinstance(module, old_module):
            hooks.append(module.register_forward_hook(make_hook(name))
            )

    # Run dummy forward pass
    model(dummy_input)

    # Clean up hooks
    for hook in hooks:
        hook.remove()

    # Replace modules
    for name, module in model.named_modules():
        if isinstance(module, old_module) and name in module_metadata:
            num_input_dims, out_channels = module_metadata[name]
            ct = new_module(num_input_dims=num_input_dims,
            out_channels=out_channels, **kwargs).to(device)

            # Replace module in the model
            names = name.split(".")
            parent = model
            for n in names[:-1]:
                if n.isdigit():
                    parent = parent[int(n)]  # for Sequential/
                    ModuleList
                else:
                    parent = getattr(parent, n)

            last_name = names[-1]
            if last_name.isdigit():
                parent[int(last_name)] = ct  # for Sequential/
                ModuleList
            else:
                setattr(parent, last_name, ct)

    return model
```

# E  LoRA Implementation

The following code provides the Python implementation of LoRA used in Section 4:

- `LoRALinear` & `LoRAConv2d`: classes that define LoRA-enhanced versions of the `Linear` and `Conv2d` modules.
- `get_lora_model`: a function that replaces all `Linear` and `Conv2d` modules in a model with their corresponding LoRA versions.

```python
import torch
from torch import nn as nn
from torch.nn import functional as F

class LoRALinear(nn.Module):
    """
    A Linear layer that applies LoRA to a frozen, pretrained Linear.
    """

    def __init__(self, original_layer: nn.Linear, r: int = 4, alpha:
    float = 1.0):
        super().__init__()
        self.in_features = original_layer.in_features
        self.out_features = original_layer.out_features
        self.r = r
        self.alpha = alpha

        # Freeze the original layer's parameters
        self.weight = nn.Parameter(original_layer.weight.data,
        requires_grad=False)
        if original_layer.bias is not None:
            self.bias = nn.Parameter(original_layer.bias.data,
            requires_grad=False)
        else:
            self.bias = None

        # LoRA parameters B and A
        # B: [out_features, r]
        # A: [r, in_features]
        self.B = nn.Parameter(torch.zeros((self.out_features, r)))
        self.A = nn.Parameter(torch.zeros((r, self.in_features)))

        # Initialize LoRA weights
        nn.init.kaiming_uniform_(self.B, a=5 ** 0.5)
        nn.init.zeros_(self.A)

    def forward(self, x):
        # Normal forward with the frozen weight
        result = F.linear(x, self.weight, self.bias)

        # LoRA path: B @ A
        # shape of BA = [out_features, in_features]
        # Then F.linear with BA
        lora_update = F.linear(x, (self.alpha / self.r) * (self.B @
        self.A))

        return result + lora_update
```

```python
class LoRAConv2d(nn.Module):
    """
    A Conv2d layer that applies LoRA to a frozen, pretrained Conv2d.
    """

    def __init__(self, original_layer: nn.Conv2d, r: int = 4, alpha:
    float = 1.0):
        super().__init__()

        self.out_channels = original_layer.out_channels
        self.in_channels = original_layer.in_channels
        self.kernel_size = original_layer.kernel_size
        self.stride = original_layer.stride
        self.padding = original_layer.padding
        self.dilation = original_layer.dilation
        self.groups = original_layer.groups
        self.bias_available = (original_layer.bias is not None)

        self.r = r
        self.alpha = alpha

        # Freeze original parameters
        self.weight = nn.Parameter(original_layer.weight.data,
        requires_grad=False)
        if self.bias_available:
            self.bias = nn.Parameter(original_layer.bias.data,
            requires_grad=False)
        else:
            self.bias = None

        # Flattened shape for weight is [out_channels, in_channels *
        k_h * k_w]
        k_h, k_w = self.kernel_size
        fan_in = self.in_channels * k_h * k_w  # Flattened input dim

        # Define LoRA parameters: B and A
        # B: [out_channels, r]
        # A: [r, fan_in]
        self.B = nn.Parameter(torch.zeros((self.out_channels, r)))
        self.A = nn.Parameter(torch.zeros((r, fan_in)))

        # Initialize LoRA weights
        nn.init.kaiming_uniform_(self.B, a=5 ** 0.5)
        nn.init.zeros_(self.A)

    def forward(self, x):
        # Standard (frozen) convolution
        result = F.conv2d(
            x,
            self.weight,
            bias=self.bias,
            stride=self.stride,
            padding=self.padding,
            dilation=self.dilation,
            groups=self.groups
        )

        # Compute LoRA update
        # 1) Flatten conv kernel in the same manner as above
        # 2) Multiply B and A -> shape [out_channels, in_channels *
        k_h * k_w]
        # 3) Reshape it back to [out_channels, in_channels, k_h, k_w]
        BA = self.B @ self.A  # shape [out_channels, fan_in]
```

```python
        # Reshape to conv kernel
        k_h, k_w = self.kernel_size
        lora_weight = BA.view(
            self.out_channels,
            self.in_channels,
            k_h,
            k_w
        ) * (self.alpha / self.r)

        # Perform conv2d with the LoRA weight (no extra bias term for
        LoRA)
        lora_update = F.conv2d(
            x,
            lora_weight,
            bias=None,
            stride=self.stride,
            padding=self.padding,
            dilation=self.dilation,
            groups=self.groups
        )

        return result + lora_update
```

```python
def get_lora_model(model: nn.Module, r: int = 4, alpha: float = 1.0):
    """
    Recursively replace all Conv2d and Linear modules in model with
    LoRA-enabled versions. Freezes original weights and adds LoRA
    parameters.
    """
    for name, child in list(model.named_children()):
        # If child is a Conv2d, replace it with LoRAConv2d
        if isinstance(child, nn.Conv2d):
            lora_module = LoRAConv2d(child, r=r, alpha=alpha)
            setattr(model, name, lora_module)

        # If child is a Linear, replace it with LoRALinear
        elif isinstance(child, nn.Linear):
            lora_module = LoRALinear(child, r=r, alpha=alpha)
            setattr(model, name, lora_module)

        else:
            # Recursively traverse children
            get_lora_model(child, r=r, alpha=alpha)

    return model
```

