# OpenReview forum: "Curvature Tuning: Provable Training-free Model Steering From a Single Parameter"
_NeurIPS.cc/2025/Workshop/Reliable_ML — NeurIPS 2025 - Reliable ML Workshop_

### Official Review · Reviewer_8H8a · 2025-09-19

**Rating:** 6
**Confidence:** 4

**Review:**

This paper proposes Curvature Tuning (CT), a method to steer pretrained models by injecting a scalar parameter $\beta$ into every activation to smoothly modulate decision boundary curvature without weight updates. Built on a spline-theoretic view of ReLU networks, CT replaces ReLUs with **CT Units (CTUs)** that interpolate between linear and nonlinear regimes via $\beta$. As $\beta\ \downarrow$, the model becomes smoother (lower curvature). The paper provides empirical results showing that CT improves accuracy and adversarial robustness, while Trainable CT (learnable per-neuron $\beta,c$) outperforms LoRA on many tasks using ≤ numbers of LoRA’s parameters.

---

## Strengths

- **Good Presentation:** Paper is well structured, with good figures, reproducibility details, and clear theoretical-experimental linkage. I like that the main paper is short and is pretty well suited to a workshop. The Appendix is well-detailed (in full disclosure, I haven't gone through its entirety).
- **Sufficiently Novel:** Introduces a principled, activation-centric fine-tuning method grounded in the spline interpretation of ReLU networks. This offers an interpretable knob to control model curvature.  The authors leverage known results from spline theory to justify their approach, and they derive the CTU activation by combining two smooth approximations of a max operator (soft region selection via entropy regularization and log-sum-exp smoothing) – a principled construction that leads to a family of activations encompassing ReLU, SoftPlus, SiLU, even GELU - Which I like a lot.
- **Strong results**: On 12 image tasks, even training-free CT improves over linear probing; Trainable CT exceeds LoRA-r1 despite fewer parameters. Also shows 10–15× gains in $\ell_\infty$ adversarial robustness with no adversarial training.  These results, combined with the code snippet in appendix, are enough to establish technical correctness.
- **Sound theory:** Provides proofs that CT yields smoother functions with bounded derivatives, showing how $\beta$ directly controls decision-boundary curvature (as $\beta\ \downarrow$, curvature $\to 0$).

---

## Weaknesses
- **Baselines are limited:** Only LoRA-r1 is used; stronger baselines (LoRA-r4, full fine-tuning) would clarify CT’s absolute performance. I understand it's not a direct comparison since CT would have much fewer parameters. But in general, I would be interested to see these results.
- **Scope is narrow:** Validated only on CNNs for vision; unclear if it generalizes to other domains.
- **Interpretability diluted in Trainable CT:** Once thousands of $(\beta,c)$ are learned, it becomes less interpretable than the single-$\beta$ steering mode.

---

## Questions for Authors
1. How does Trainable CT perform vs LoRA with higher rank (r=4/8) or full fine-tuning?
2. What patterns do the learned $\beta_i,c_i$ show across layers?
3. Any measured inference overhead from CTUs?
4. Have you tried CT on transformer/LN-heavy or NLP models?

---

In general, the paper is clear, original, has as well-executed idea with strong theory and results; especially impressive parameter-efficiency and robustness.  I would recommend the paper to be accepted but would also like to see some more results (see questions) for the appendix.

---

### Official Review · Reviewer_2ssq · 2025-09-20
**Review for paper: Curvature Tuning: Provable Training-free Model Steering From a Single Parameter**

**Rating:** 7
**Confidence:** 3

**Review:**

# Summary

## Claims

The paper proposes Curvature Tuning (CT), a simple, interpretable way to steer pretrained models by changing the activation functions rather than the weights. A single scalar $\beta \in [0,1]$ in injected into all activations to control the curvature.

## How it does it

Authors soften (i) region selection via an entropy-regularized argmax and (ii) the maximum function via log-sum-exp, and combine these into a parametric activation, the CT Unit (CTU) with one parameter $c \in [0,1]$ for mixing coefficient.

## Main results

The paper has mainly two lines of results:

- **Generalization (12 downstream datasets on ResNet-18/50/152)**: CT beats linear probing, having relative gains of 6.75%/8.59%/8.34%. It also outperforms LoRA while training far fewer parameters.
- **Robustness (RobustBench $l_2 / l_\infty /$corruption)**: CT improves robustness without adversarial training, with particularly large relative gains for $l_\infty$ attacks.

# Strengths

1. The global scalar $\beta$ is simple and interpretable. It governs the nonlinearity smoothness, and it's easy to implement.
2. The training process is almost zero-cost: no backprop, no weight updates. For the experiment in the paper (small scales), it works well. Also it has good parameter efficiency, even much fewer than LoRA.

# Weaknesses

Maybe more experiments can be conducted to see the generalizability of this approach (more baselines methods, more larger models, more domains, not only vision-CNN-centric experiments etc.)

# Suggestions for Authors & Ethics

None